# Effect of Hydroxybenzoic Acids on Caffeine Detection Using Taste Sensor with Lipid/Polymer Membranes

**DOI:** 10.3390/s22041607

**Published:** 2022-02-18

**Authors:** Zeyu Zhao, Misaki Ishida, Takeshi Onodera, Kiyoshi Toko

**Affiliations:** 1Graduate School of Information Science and Electrical Engineering, Kyushu University, 744 Motooka, Nishi-ku, Fukuoka 819-0395, Japan; ishida.misaki.812@s.kyushu-u.ac.jp (M.I.); onodera@ed.kyushu-u.ac.jp (T.O.); 2Institute for Advanced Study, Kyushu University, 744 Motooka, Nishi-ku, Fukuoka 819-0395, Japan; toko@ed.kyushu-u.ac.jp; 3Research and Development Center for Five-Sense Devices, Kyushu University, 744 Motooka, Nishi-ku, Fukuoka 819-0395, Japan

**Keywords:** taste sensor, lipid/polymer membrane, hydroxybenzoic acids, caffeine detection

## Abstract

A taste sensor with lipid/polymer membranes can objectively evaluate taste. As previously reported, caffeine can be detected electrically using lipid/polymer membranes modified with hydroxybenzoic acids (HBAs). However, a systematic understanding of how HBAs contribute to caffeine detection is still lacking. In this study, we used various HBAs such as 2,6–dihydroxybenzoic acid (2,6–DHBA) to modify lipid/polymer membranes, and we detected caffeine using a taste sensor with the modified membranes. The effect of the concentrations of the HBAs on caffeine detection was also discussed. The results of the caffeine detection indicated that the response to caffeine and the reference potential measured in a reference solution were affected by the log P and pKa of HBAs. Furthermore, the taste sensor displayed high sensitivity to caffeine when the reference potential was adjusted to an appropriate range by modification with 2,6–DHBA, where the slope of the change in reference potential with increasing 2,6–DHBA concentration was steep. This is helpful in order to improve the sensitivity of taste sensors to other taste substances, such as theophylline and theobromine, in the future.

## 1. Introduction

Generally, there are five basic tastes sensed by humans, namely, sourness, bitterness, sweetness, saltiness, and umami. In the past, human sensory evaluation was used to evaluate the taste quality of products [1]. However, this method is highly subjective owing to the differences in physiological and psychological conditions among evaluators. Thus, an objective and quantitative method of evaluating taste was required.

For this reason, technologies that can quantify taste have been developed worldwide. Electronic tongues and taste sensors can now evaluate tastes objectively [2,3,4,5,6,7,8,9,10,11]. Electronic tongues use ion-specific electrodes [7] or pulse voltammetry [8] to provide taste information of samples using multivariate analysis or artificial neural networks. Taste sensors with lipid/polymer membranes measure taste by detecting the change in membrane potential from taste substances, and these exhibit global selectivity [9]. Taste sensors are designed to distinguish and quantify each taste in near replication of the human sense of taste. Under the premise of distinguishing and quantifying tastes, by changing the composition of its lipid/polymer membrane, a taste sensor can quantify the five basic and astringent tastes [10,11]. Furthermore, it can detect inhibitory effects appearing in coexistent sweetness and bitterness [12]. Nevertheless, taste sensors show insensitivity to non-charged taste substances because their functionality is contingent upon membrane potential measurement for charged taste substances.

In our previous study [13], non-charged bitter substances (e.g., caffeine, theophylline, and theobromine) were measured using taste sensors with lipid/polymer membrane modified with HBAs, and this method is similar to those which improve the sensitivity of sweet taste sensors to sugar by surface modification [14,15]. In the literature [13], caffeine detection has been discussed as a change in membrane potential due to allostery caused by intermolecular hydrogen bonds between HBAs and caffeine. Moreover, the sensitivity of taste sensors to caffeine detection has been related to the number of intramolecular hydrogen bonds of HBAs. However, a systematic understanding of how HBAs contribute to caffeine detection is still lacking.

HBAs are composed of three parts: the hydroxyl group, the carboxyl group, and the benzene ring. The presence of the carboxyl group allows HBAs to ionize H^+^ in solution, causing ionized HBAs to be negative. In order to describe the ionization mathematically, researchers have used the pKa value of HBAs [16,17]. When the carboxyl and hydroxyl groups are adjacent to each other on the benzene ring, they form an intramolecular hydrogen bond, which can affect the ionization of HBAs [18,19,20]. The presence of the benzene ring makes HBAs hydrophobic, which can be described by their log P value [21,22]. A cation-π interaction allows HBAs to bind cations [23,24,25]. A dimer can be formed between two HBAs through their carboxyl groups [26,27,28]. Therefore, HBAs are capable of adsorbing to a lipid/polymer membrane owing to the electrostatic and hydrophobic interactions; there is a clear need to consider the effects of cation-π interaction and dimerization of HBAs on caffeine detection.

It was previously observed that the relative magnitudes of *p*–HBA (PHBA)–caffeine pair interactions vary systematically with the concentration ratio in methanol solutions of PHBA/caffeine [29]. This indicated that, when the concentration of PHBA in the crystallizing solution increased, there was a greater likelihood of producing crystals with higher PHBA content (e.g., caffeine–2 PHBA or PHBA–PHBA crystals). Moreover, there is indirect evidence to suggest that modelling the shape of concentration-effect curves is a prime method to study the interaction between drugs and receptors in pharmacology [30,31]. Using the geometric description of the concentration-effect curve (e.g., the mid-point slope), the change in agonist function after a system modification was assessed. These studies have revealed a need for discussing the effect of HBA concentration on caffeine detection.

In this study, we used the surface modification method by immersing the sensor electrode in a modification solution that contained six types of HBA (2,6–DHBA, 3,4–dihydroxybenzoic acid, 3,5–dihydroxybenzoic acid, 2–hydroxybenzoic acid, 2,5–dihydroxybenzoic acid, and 2,4,6–trihydroxy benzoic acid). We compared the results obtained with and without surface modification using various HBAs, and the effects of HBAs on caffeine detection were evaluated. To investigate the effect of the HBA concentration on caffeine detection, we detected caffeine using a lipid/polymer membrane modified with HBAs of various concentrations, and then we plotted HBA–concentration-effect curves, whose vertical coordinate is the reference potential or the response to caffeine while the horizontal coordinate is the HBA concentration. The reference potential *Vr* is measured in a reference solution before detecting a caffeine sample. By comparing it with various HBA-concentration-effect curves for caffeine detection, the effect of HBA concentration on caffeine detection is evaluated. The results of caffeine detection revealed that the reference potential and the response to caffeine were affected by the log P and pKa of HBAs. In addition, when the slope of the change in *Vr* with increasing 2,6–DHBA concentration is steep, the membrane potential can be changed more efficiently, resulting in the taste sensor being more sensitive to caffeine.

## 2. Materials and Methods

### 2.1. Reagents

The lipid used for making lipid/polymer membrane was tetradodecylammonium bromide (TDAB), and the preparation solvent was tetrahydrofuran (THF), both of which were purchased from Sigma-Aldrich, Inc. (St. Louis, MO, USA). Dioctyl phenyl-phosphonate (DOPP) and polyvinyl chloride (PVC) were purchased from FUJIFILM Wako Pure Chemical Corporation (Osaka, Japan). The caffeine sample was purchased from Tokyo Chemical Industry Co., Ltd. (Tokyo, Japan). The modifiers studied were 2,6–DHBA, 3,4-dihydroxybenzoic acid (3,4–DHBA), 3,5–dihydroxybenzoic acid (3,5–DHBA), 2–hydroxybenzoic acid (2–HBA), 2,5–dihydroxybenzoic acid (2,5–DHBA), and 2,4,6–trihydroxy benzoic acid (2,4,6–THBA). All modifiers were purchased from Kanto Chemical Co., Inc. (Tokyo, Japan). Tannic acid and potassium chloride (KCl) were used to make the reference solution, and these were purchased from Kanto Chemical Co., Inc. (Tokyo, Japan). Figure 1 shows the molecular formulae of TDAB, DOPP, and caffeine.

### 2.2. Lipid/Polymer Membrane

The lipid/polymer membrane was composed of lipid, plasticizer, and PVC. The hydrophobicity of the membrane was adjusted by the lipid and plasticizer. The electricity of the membrane was adjusted by the lipid, and the softness of the membrane was adjusted by the plasticizer. PVC was the support agent for the membrane.

The lipid/polymer membrane was formed by mixing 10 mL of 0.3, 1, or 3 mM TDAB in THF as a lipid, 1.5 mL DOPP as a plasticizer, and 800 mg PVC as a support agent. The mixture solution was poured into a Petri dish (90 mm φ), and the membrane was formed by volatilizing the THF, and then a piece of the membrane was pasted on a probe (Figure 2).

### 2.3. Measurement Procedure of Taste Sensor

All measurements in this study were based on the commercialized TS–5000Z taste sensing system (Intelligent Sensor Technology, Inc., Kanagawa, Japan). The detection unit of the taste sensor is a device that can be equipped with a reference electrode and up to eight sensor electrodes. As shown in Figure 3a, the structure of this sensor electrode consisted of a probe with a hole, whose surface can adhere to a lipid/polymer membrane. The inside of the sensor electrode was filled with 200 μL of saturated AgCl/3.33 M KCl solution. For the reference electrode, an Ag/AgCl-saturated KCl reference electrode was used. The change in membrane potential between the reference and sensor electrodes was used as the sensor output.

Figure 3b shows the measurement procedure of the taste sensor. Firstly, the sensor and reference electrodes were immersed in a reference solution composed of 30 mM KCl and 0.3 mM tartaric acid. The membrane potential *Vr* was measured for 30 s. This *Vr* is an important parameter of the taste sensor because it represents the measurement environment before measuring the sample solution, and it is helpful for evaluating the membrane surface charge density [32]. Secondly, the sensor and reference electrodes were immersed in a sample solution, and the membrane potential *Vs* was measured for 30 s. The difference between *Vs* and *Vr* was considered as the response value. Thirdly, the membrane potential *Vr’* caused by adsorption was measured. The difference between *Vr’* and *Vr* was considered as the CPA (change of membrane potential caused by adsorption) value, which can provide data to evaluate the aftertaste caused by the strong adsorption of taste substances on the human tongue [10]. Finally, the membrane surface was refreshed with a solution of 10 mM KOH, 100 mM KCl, and 30 vol% EtOH. The entire process was performed three to five times. The mean values and standard deviations (SDs) were calculated from n = 4 (electrode) × 4 (rotation) = 16 electrical response values using the same method as in [33].

### 2.4. Modification of Lipid/Polymer Membrane

Our previous study [14] demonstrated that the sensitivity of the taste sensor to taste substances was improved by soaking the lipid/polymer membrane in a modification solution. As reported by [34], liquid-membrane electrodes with electrically charged ion-exchange sites generally exhibit permselectivity for oppositely charged counterions, which indicated that liquid-membrane electrodes with TDAB membranes have a strong electrostatic interaction on negatively charged ions. Due to the fact that the Br^−^ from TDAB and the ionized HBAs have the same negative charge, an ion exchanger can take place on the membrane surface. However, because of the overwhelming hydrophobicity of HBA, ionized HBAs can be adsorbed onto the membrane surface by hydrophobic interaction. Therefore, the lipid/polymer membrane was modified by immersing the sensor electrode with the lipid/polymer membrane in a solution containing HBAs for 72 h.

### 2.5. Mechanism of Caffeine Detection Using Taste Sensor with Lipid/Polymer Membrane Modified with HBAs

In the study [35], the interactions between caffeine and HBA were formed by intermolecular hydrogen bonds (e.g., the O-H(carboxy) N(imidazole) hydrogen bond). On the basis of this result, the study [13] has discussed the mechanism of caffeine detection as the following: owing to intermolecular hydrogen bonds between HBAs and caffeine, the carboxyl groups of HBAs removed H^+^ from the caffeine solution, which caused allostery. The allostery resulted in a change in membrane potential, which is used as the response to caffeine. The change in membrane potential through this mechanism reflected the effect of HBAs on caffeine detection. In conjunction with the measurement procedure of the taste sensor, we considered three membrane potentials (*Vr*, *Vs–Vr*, and *Vr’–Vr*) as the factors affected by HBAs.

### 2.6. Effect of HBAs on Caffeine Detection Using Taste Sensors

To confirm the effect of HBAs on caffeine detection, we used six types of HBA to modify the lipid/polymer membrane (Figure 4). Caffeine in the reference solution was detected by the taste sensor with the modified lipid/polymer. The compositions and concentrations of the membranes and HBAs are shown in Table 1.

### 2.7. Effect of HBA Concentration on Caffeine Detection

To confirm the effect of HBA concentration on caffeine detection, as the next step, we used HBAs with six different concentrations in order to modify the lipid/polymer membrane (Table 2). Caffeine in the reference solution was measured by the taste sensor with the modified lipid/polymer membrane as described in Section 2.6.

## 3. Results

### 3.1. Measurement of Caffeine by Surface Modification Method Using Various HBAs

#### 3.1.1. Effects of Log P and pKa of HBA on Reference Potential

Figure 5 shows the reference potential (*Vr*) for 100 mM caffeine measured by the taste sensor with the lipid/polymer membrane modified with six types of HBA, in which the concentration of HBAs in the modification solution is the same as mentioned in Table 1. As shown by the results, compared with *Vr* obtained without surface modification, *Vr* for 2,6–DHBA, 2,5–DHBA, 2,4,6–THBA, 2–HBA, and 3,4–DHBA/0.3 mM TDAB was low; and *Vr* for 3,5–DHBA, 3,4–DHBA/1 mM TDAB, and 3,4–DHBA/3 mM TDAB was high. In particular, the membranes modified with 2,6–DHBA and 2–HBA showed relatively low *Vr* values; the membranes modified with 2,4,6–DHBA and 2,5–DHBA showed relatively high *Vr* values; and the 1, 3 mM TDAB membranes modified with 3,5–DHBA and 3,4–DHBA showed high *Vr* values. Table 3 gives the pKa values calculated from Marvin (Marvin 21.12.0, ChemAxon, Budapest, Hungary) and the log P values calculated from ACD/Labs Release (12.00, version 12.01, Advanced Chemistry Development, Inc., Toronto, ON, Canada) for the HBAs. The pKa of 2,6–DHBA was the smallest among the HBAs, which indicated that the ionization of 2,6–DHBA was the greatest. The Log P of 2,6–DHBA was the largest, which indicated that the hydrophobicity of 2,6–DHBA was the highest.

#### 3.1.2. Effects of log P and pKa of HBA on Response to Caffeine

Figure 6 shows the response to 100 mM caffeine measured by the taste sensor with the lipid/polymer membrane modified with six types of HBA, where the concentration of HBAs in all modification solutions was 0.1 wt%. The response to caffeine increased after surface modification with the HBAs. In particular, the membranes modified with 2,6–DHBA showed the greatest response to caffeine. A moderate response appeared in the membranes modified with 2,4,6–THBA, 2–HBA, and 2,5–DHBA. The smallest response was obtained for the membranes modified with 3,4–DHBA and 3,5–DHBA.

### 3.2. Effect of HBA Concentration on Caffeine Detection Using Taste Sensors

#### 3.2.1. 2,6–DHBA

As reported by [13], the sensitivity of taste sensors to caffeine detection has been related to the number of intramolecular hydrogen bonds of HBAs. Moreover, as shown by Figure 6, the membrane modified with 2,6–DHBA responds best to caffeine. By comparing 2,6–DHBA with other HBAs, the best response conditions for caffeine detection can be derived. Therefore, the data for 2,6–DHBA with two intramolecular hydrogen bonds were selected to be studied in this section.

A total of 100 mM caffeine in the reference solution was detected by the taste sensor with lipid/polymer membranes modified with various concentrations of 2,6–DHBA. The results were used to compose 2,6–DHBA–concentration-effect curves (E/[2,6–DHBA] curves, where E is the abbreviation for “effect” and [2,6–DHBA] is the 2,6–DHBA concentration), which were depicted on a semi-logarithmic scale (Figure 7). Figure 7a shows three E/[2,6–DHBA] curves for *Vr*. For the 0.3 mM TDAB membrane, as the 2,6–DHBA concentration increased, *Vr* decreased and then increased at 0.03 wt% 2,6–DHBA. For the 1, and 3 mM TDAB membranes, *Vr* decreased with increasing 2,6–DHBA concentration and then increased at 0.1 wt% 2,6–DHBA. Figure 7b shows the three E/[2,6–DHBA] curves for the response to caffeine. For the 0.3, and 1 mM TDAB membranes, the response to caffeine increased with increasing 2,6–DHBA concentration and showed a peak at a high 2,6–DHBA concentration. For the 3 mM TDAB membrane, as the 2,6–DHBA concentration increased, the response to caffeine increased and then plateaued at a high 2,6–DHBA concentration. This indicated that using 0.3 wt% 2,6–DHBA to modify the lipid/polymer membrane with 3 mM TDAB effectively enhanced the response to caffeine.

Moreover, we measured the membrane potential (*Vr’*) in the reference solution after the caffeine sample detection, and then we considered the difference between *Vr’* and *Vr* as the CPA value. Figure 7c shows three E/[2,6–DHBA] curves for the CPA value. For the 0.3, and 1 mM TDAB membranes, CPA showed a peak with increasing 2,6–DHBA concentration; for the 3 mM TDAB membrane, the CPA value increased and then plateaued at 0.03 wt% 2,6–DHBA.

#### 3.2.2. 2–HBA

In our previous study [13], the data for 2–HBA with one intramolecular hydrogen bond were not studied. Therefore, the data for 2–HBA was selected to be investigated in this section.

Figure 8 shows 2–HBA-concentration-effect (E/[2–HBA]) curves for caffeine detection. As shown in Figure 8a, for the 0.3 and 3 mM TDAB membranes, *Vr* decreased with increasing 2–HBA concentration, and then increased when the 2–HBA concentration exceeded 0.1 wt%; for the 1 mM TDAB membrane, as the 2-HBA concentration increased, Vr decreased and then plateaued at 0.1 wt% 2–HBA. Figure 8b shows the relationship between the 2–HBA concentration and the response to caffeine. The response to caffeine increased with the 2–HBA concentration. This indicated that the sensitivity of the lipid/polymer membrane to caffeine could be improved by increasing 2–HBA concentration. However, the effect is less than that of 2,6–DHBA. As shown in Figure 8c, for the 1 and 3 mM TDAB membranes, the CPA value increased with the 2–HBA concentration; for the 0.3 mM TDAB membrane, the CPA value increased with increasing 2–HBA concentration, and then decreased at 0.03 wt% 2–HBA.

#### 3.2.3. 3,5–DHBA

The 3,5–DHBA without intramolecular hydrogen bond was not studied in our previous study [13]. Therefore, the data for 3,5–DHBA were investigated in this section.

Figure 9 shows 3,5–DHBA-concentration-effect (E/[3,5–DHBA]) curves for caffeine detection. As shown in Figure 9a, for the 1 and 3 mM TDAB membranes, *Vr* increased with increasing 3,5–DHBA concentration. For the 0.3 mM TDAB membrane, *Vr* decreased with increasing 3,5–DHBA concentration, and then increased when the 3,5–DHBA concentration exceeded 0.01 wt%. As shown in Figure 9b, although the response to caffeine increased with the 3,5–DHBA concentration, all response values were small and could be negligible. The negligible response to caffeine verified the result in Section 3.1, in which Figure 6 shows an inadequate response to caffeine for the membranes modified with 3,5–DHBA. As shown in Figure 9c, although CPA increased with the 3,5–DHBA concentration, the CPA value of the 3,5–DHBA modified membrane was small (<4 mV).

## 4. Discussion

### 4.1. Comparison of Reference Potentials among HBAs

Comparing the results obtained with and without surface modification using various HBAs, the effects of HBAs on the reference potential were evaluated. Firstly, from Figure 5, the results indicate that the reference potential varied with the surface modification of HBAs. The ionization of HBAs can be used to interpret this phenomenon. The ionized HBAs were adsorbed to the lipid/polymer membrane through the electrostatic and hydrophobic interactions, and then they neutralized the positive charge of the lipid TDAB, causing the surface charge density of the membrane to decrease, resulting in the *Vr* being smaller than that obtained without surface modification.

In the second place, Figure 5 and Table 3 together indicate that, at the same pKa level (such as 2,6–DHBA and 2,4,6–THBA; 2,5–DHBA and 2–HBA), an HBA with a larger log P resulted in a smaller *Vr*. For example, the log P of 2,6–DHBA is 2.24, which is higher than that of 2,4,6–THBA (1.80). When they were used to modify the 3 mM TDAB membranes respectively, the *Vr* of the 2,6–DHBA-modified membrane was 53.36 mV, which was smaller than the *Vr* of the 2,4,6–THBA-modified membranes (=107.70 mV). Similarly, the log P of 2–HBA (=2.06) was higher than the log P of 2,5–DHBA (=1.56), and the *Vr* of the 2–HBA-modified membrane (42.71 mV) was smaller than that of the 2,5–DHBA-modified membrane (93.56 mV). These results can be interpreted in terms of the hydrophobicity of the HBAs: an HBA with a higher hydrophobicity is more likely to adsorb on the lipid/polymer membrane, resulting in a smaller reference potential.

In addition, for 3,5–DHBA and 3,4–DHBA, with high or relatively high pKa, the ionization of these is so small that they cannot affect the positive charge of lipid TDAB, although cations can be bound to the membrane owing to the cation−π interaction. These phenomena resulted in the *Vr* for the 3,5–DHBA or 3,4–DHBA-modified membrane being higher than those obtained without surface modification. Nevertheless, for the membrane with less lipid TDAB, because the surface charge density of this membrane is small, it can be affected by 3,4–DHBA with relatively high pKa, resulting in the *Vr* for 3,4–DHBA-modified 0.3 mM TDAB membrane being smaller than that obtained without surface modification.

### 4.2. Comparison of Responses to Caffeine among HBAs

Comparing the results obtained with and without surface modification using various HBAs, the effects of HBAs on the response to caffeine were evaluated. From Figure 6, surface modification with HBAs improved the sensitivity of the lipid/polymer membrane to caffeine. Combining this observation with Table 3, it can be observed that the sensitivity of the taste sensor to caffeine varied with the pKa of the HBAs. For 2,6–DHBA and 2–HBA at the same log P level, owing to the pKa of 2,6–DHBA being smaller than that of 2–HBA, the more ionized H^+^ can be removed to the carboxyl group of 2,6–DHBA from the caffeine solution through the formation of intermolecular hydrogen bonds between 2,6–DHBA and caffeine, resulting in the 2,6–DHBA-modified membrane being sensitive to caffeine. Similarly, for the same log P level of HBAs such as 2,4,6–THBA and 2,5–DHBA; 3,4–DHBA and 3,5–DHBA, a membrane modified with an HBA containing a smaller pKa showed a greater response to caffeine. Moreover, for 2,6–DHBA and 2,4,6–THBA, with the same level of pKa, the log P of 2,6–DHBA was higher than that of 2,4,6–THBA, indicating that 2,6–DHBA was more likely to be adsorbed to the membrane, causing the adsorption of 2,6–DHBA to be more than that of 2,4,6–THBA, resulting in more ionized H^+^ that can be removed to the membrane. These phenomena revealed that the greater adsorption and ionization of HBAs, the more H^+^ can be removed from caffeine sample solution, and, consequently, the greater the change in membrane potential detected by the taste sensor, resulting in a higher sensitivity of taste sensor to caffeine.

### 4.3. Comparison of CPA Values among HBAs

As can be seen from Figure 7c for 2,6–DHBA, Figure 8c for 2–HBA, and Figure 9c for 3,5–DHBA, the trends of the CPA values and the response to caffeine were consistent, which indicated that the effects of the HBAs on CPA value and the response to caffeine were the same. The CPA value was generated when taste substances were adsorbed onto the surface of the lipid/polymer membrane, changing the membrane surface charge density [11,36,37,38]. The distribution coefficient (log D) of caffeine is around −0.44 at pH = 3.5 (reference solution pH), which indicates that the adsorption of caffeine did not require the hydrophobic interaction. Moreover, caffeine could not be adsorbed into the membranes by the electrostatic interaction because it cannot be ionized in solution. This observation, coupled with the consistency of the trends between the CPA value and the caffeine response, indicated that the main factor of caffeine adsorption was intermolecular hydrogen bonds between HBAs and caffeine. It was previously revealed that caffeine can systematically form co-crystals with HBAs via both O-H(hydroxy) O(urea), O-H(hydroxy) O(amide), and O-H(carboxy) N(imidazole) hydrogen bonds [39,40,41,42].

### 4.4. Effect of HBA Concentration on Reference Potential

Figure 7a, Figure 8a and Figure 9a show the relationship between the reference potential and the HBA concentration: for 2,6–DHBA and 2–HBA, *Vr* showed a negative peak with increasing 2,6–DHBA or 2–HBA concentration; for 3,5–DHBA, *Vr* increased with the 3,5–DHBA concentration. This relationship can be interpreted by considering two complementarities: (i): the negative charge of the ionized HBA, and the positive charge of the lipid TDAB; and (ii): the negative charge of the ionized HBA, and the cations bound by the benzene ring of the HBAs through the cation-π interaction. At a low 2,6–DHBA or 2–HBA concentration, the cation-π interaction effect is still negligible. The negative charge of ionized 2,6–DHBA or 2–HBA is sufficiently strong to neutralize the positive charge of the lipid TDAB, causing the surface charge density of the membrane to decrease, resulting in a decrease in *Vr*. At a higher 2,6–DHBA or 2–HBA concentration, e.g., 0.1 wt% in the experiment, the cation-π interaction is no longer negligible, resulting in an increase in *Vr*. As shown in Table 3, the pKa of 3,5–DHBA was 4.16, and the log P was 1.16 ± 0.24. Therefore, 3,5–DHBA has low hydrophobicity and negativity, which made it difficult for 3,5–DHBA to reduce the positive charge of the lipid TDAB, resulting in an increase in *Vr* owing to complementarity (ii).

### 4.5. Relationship between Reference Potential and Response to Caffeine

As reported in the literature [43], in the medium region of lipid concentration, a slight change in charge density can easily induce a large shift in membrane potential, resulting in high sensitivity to a bitter substance. According to [43], Figure 10 was constructed to indicate the relationship between the reference potential and the response to caffeine. As shown in Figure 10, when the slope of the change in *Vr* with increasing 2,6–DHBA concentration was steep (in the grey area of Figure 10), the response to caffeine was large. To explain this relationship, we discussed the essence of caffeine detection as follows. The formation of intermolecular hydrogen bonds between caffeine and HBA causes the ionized carboxyl groups of the HBAs to remove H^+^ from the caffeine solution, causing the surface charge density of the membrane to be close to a surface charge density of the membrane where HBA is not ionized, resulting in a positive response to caffeine.

On the basis of this essence of caffeine detection, when the concentration of 2,6–DHBA was low, there was little adsorption of 2,6–DHBA on the membrane, resulting in a small response to caffeine. When the slope of the change in *Vr* was steep (in the grey area of Figure 10), the *Vr* was more likely to be changed by the caffeine detection, resulting in a large response to caffeine. For example, as shown by the red arrow A in Figure 10a, at 0.1 wt% 2,6–DHBA, the detection of caffeine resulted in a change in surface charge density of the membrane, causing a 40-mV shift in *Vr*, which corresponds to the large response to caffeine in Figure 10b. At 0.3 wt% 2,6–DHBA, *Vr* was increased by the following. Firstly, 2,6–DHBA dimers were formed by intermolecular hydrogen bonds between the carboxyl groups, causing fewer carboxyl groups to interact with caffeine, resulting in less adsorption of caffeine (e.g., as shown by Figure 7c, for the 1 mM TDAB membrane, the CPA value for 0.3 wt% 2,6–DHBA = 4.29 mV < the CPA value for 0.1 wt% 2,6–DHBA = 5.30 mV). Secondly, the cations were bound by 2,6–DHBA owing to cation-π interaction, causing the *Vr* to increase. Owing to the dimerization of HBA, and the cation-π interaction, the surface charge density of the membrane cannot undergo a large change, resulting in the response for 0.3 wt% 2,6–DHBA being smaller than that in the grey area of Figure 10b.

Moreover, for the 2,6–DHBA-modified membrane with 0.3 mM TDAB, at a high 2,6–DHBA concentration, the response to caffeine was low. We considered that this was related to the slope of the change in *Vr*. As shown in the blue area of Figure 10c, the slope of the change in *Vr* with increasing 2,6–DHBA concentration was small, indicating the difficulty in changing *Vr*, resulting in an inadequate response to caffeine. For example, as shown by the red arrow B in Figure 10c, inducing a 40-mV shift in *Vr* required a dramatic change in the surface charge density of the membrane. However, this is impossible because of the lesser adsorption of caffeine (the CPA value for the 0.3 wt% 2,6–DHBA modified membrane with 0.3 mM TDAB = −0.03 mV). Therefore, an inadequate response to caffeine was obtained.

On the basis of the essence of caffeine detection, for 2–HBA or 3,5–DHBA modified membrane, because the slope of change in *Vr* with increasing 2–HBA or 3,5–DHBA concentration is not steep, the *Vr* for these membranes cannot be changed easily by the caffeine detection, resulting in a small positive relative value (e.g., for the 1 mM TDAB membrane, the highest response to caffeine for 2–HBA= 11.09 mV; the highest response to caffeine for 3,5–DHBA = 8.55 mV).

## 5. Conclusions

In our previous study [13], caffeine was measured using taste sensors with lipid/polymer membrane modified with HBAs. However, a systematic understanding of how HBAs contribute to caffeine detection is still lacking. In this study, we have measured caffeine using the membrane modified with six types of HBA, and we investigated the effect of the HBA concentration on caffeine detection. We found that the log P and pKa of HBA affect the reference potential and the response to caffeine: the greater the value of log P, the more likely HBA is to adsorb on the membrane, and a membrane modified with an HBA whose pKa is smaller responds more effectively to caffeine. This phenomenon revealed that HBA displays both the adsorption and detection roles: the adsorption of HBA on the membrane surface causes the negative charge of the ionized HBA to neutralize the positive charge of the lipid TDAB, resulting in a reference potential lower than that obtained without surface modification. Through the formation of intermolecular hydrogen bonds between HBA and caffeine, HBA binds caffeine on the membrane surface, and causes ionized H^+^ to be removed from caffeine solution, resulting in a change in membrane potential. In addition, upon adjusting the reference potential to an appropriate range in which the slope of the change in the reference potential was steep, the taste sensor exhibited excellent sensitivity to caffeine. This phenomenon revealed a connection between the reference potential and the response to caffeine in the taste sensor. This connection is helpful for investigating the mechanism of caffeine detection and improving the sensitivity of taste sensors to caffeine. This study therefore indicates that, by surface modification using a modifier that can form intermolecular hydrogen bonds with a non-charged taste substance, which causes allostery, a taste sensor with a membrane modified in this fashion can detect the non-charged taste substance. By utilizing this allostery, we can develop taste sensors for other non-charged taste substances in the future (e.g., theophylline and theobromine).

## Figures and Tables

**Figure 1 sensors-22-01607-f001:**
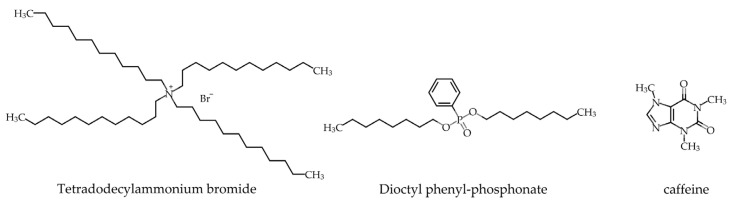
Molecular formulae of tetradodecylammonium bromide (TDAB), dioctyl phenyl-phosphonate (DOPP), and caffeine.

**Figure 2 sensors-22-01607-f002:**
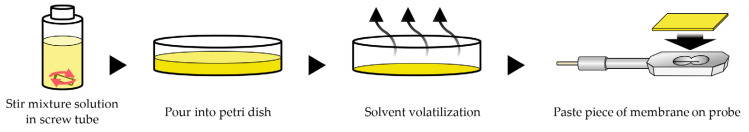
Fabrication of lipid/polymer membranes.

**Figure 3 sensors-22-01607-f003:**
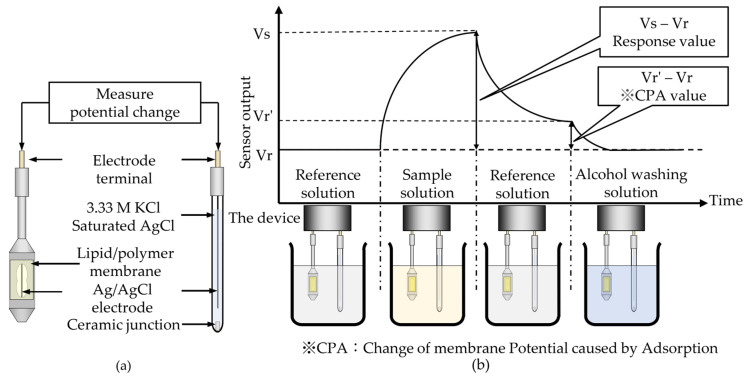
(**a**) Structure of sensor and reference electrodes. (**b**) Measurement procedure of taste sensor.

**Figure 4 sensors-22-01607-f004:**
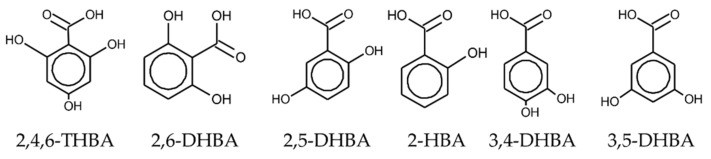
Molecular formulae of HBAs.

**Figure 5 sensors-22-01607-f005:**
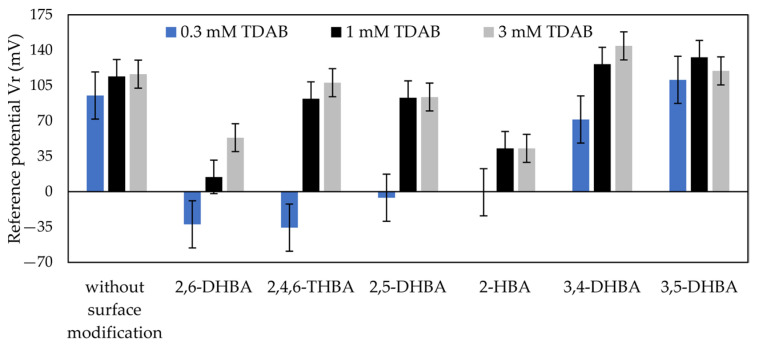
Reference potential obtained by surface modification with six types of HBAs. The error bar expresses the SD, of the data, of n = 4 (electrode) × 4 (rotation) = 16 values.

**Figure 6 sensors-22-01607-f006:**
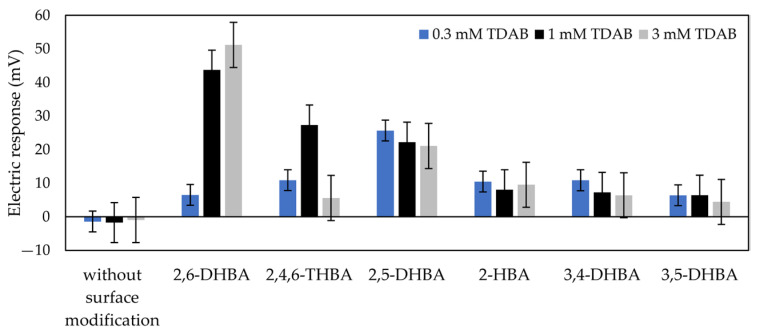
Electric response to 100 mM caffeine when using lipid/polymer membrane modified with six types of HBA. The error bar expresses the SD, of the data, of n = 4 (electrode) × 4 (rotation) = 16 values.

**Figure 7 sensors-22-01607-f007:**
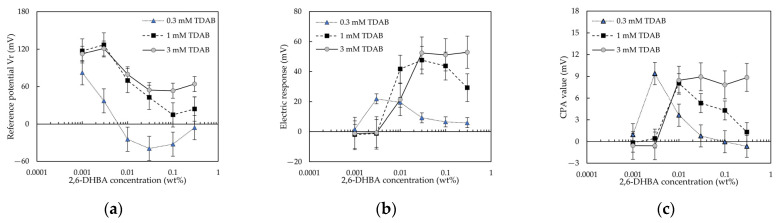
2,6–DHBA-concentration-effect curves, for 100 mM caffeine, obtained using the taste sensor with the lipid/polymer membrane (0.3, 1, 3 mM TDAB): (**a**) change in reference potential; (**b**) change in electric response; and (**c**) change in CPA value. The error bar expresses the SD, of the data, of n = 4 (electrode) × 4 (rotation) = 16 values.

**Figure 8 sensors-22-01607-f008:**
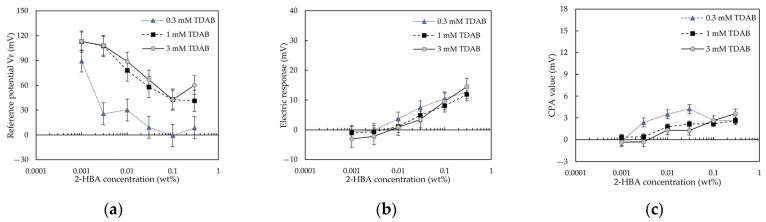
The 2–HBA-concentration-effect curves for 100 mM caffeine obtained using the taste sensor with the lipid/polymer membrane (0.3, 1, and 3 mM TDAB): (**a**) change in reference potential; (**b**) change in electric response; and (**c**) change in CPA value.

**Figure 9 sensors-22-01607-f009:**
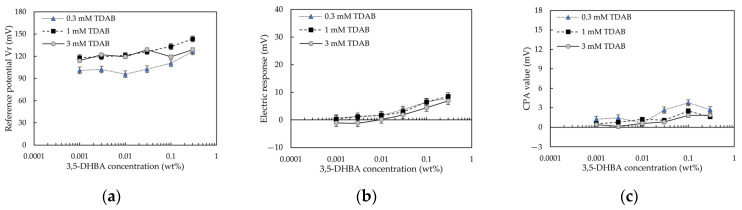
The 3,5–DHBA-concentration-effect curves for 100 mM caffeine, using the taste sensor with lipid/polymer membrane (0.3, 1, and 3 mM TDAB): (**a**) change in reference potential; (**b**) change in electric response; and (**c**) change in CPA value.

**Figure 10 sensors-22-01607-f010:**
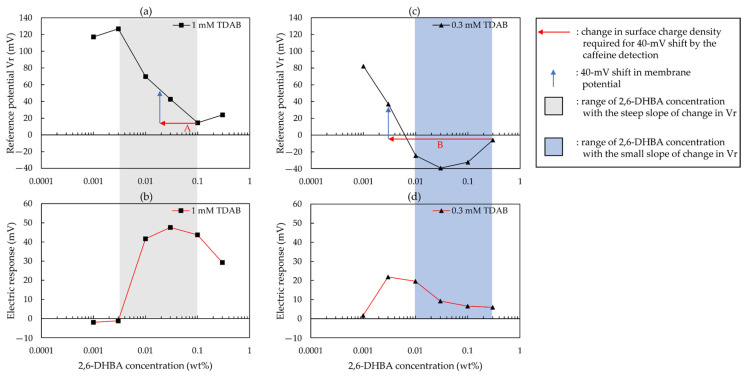
Relationship between reference potential and response to caffeine: (**a**) change in reference potential for the 1 mM TDAB membrane; (**b**) change in electric response for the 1 mM TDAB membrane; (**c**) change in reference potential for the 0.3 mM TDAB membrane; and (**d**) change in electric response for the 0.3 mM TDAB membrane.

**Table 1 sensors-22-01607-t001:** Compositions of lipid/polymer membranes and types of HBA.

Composition	Concentration
TDAB	0.3, 1, 3 mM
Caffeine	100 mM in reference solution
2,4,6–THBA, 2,6–DHBA, 2,5–DHBA, 2–HBA 3,4–DHBA, 3,5–DHBA	0.1 wt%

**Table 2 sensors-22-01607-t002:** Concentrations of modification solutions.

Composition	Concentration
2,4,6–THBA, 2,6–DHBA, 2,5–DHBA, 2–HBA 3,4–DHBA, 3,5–DHBA	0.001, 0.003, 0.01, 0.03, 0.1, 0.3 wt%

**Table 3 sensors-22-01607-t003:** The pKa and log P of HBAs.

	2,6–DHBA	2,4,6–THBA	2,5–DHBA	2–HBA	3,4–DHBA	3,5–DHBA
pKa	1.64	1.95	2.53	2.79	3.61	4.16
log P	2.24 ± 0.27	1.80 ± 0.36	1.56 ± 0.26	2.06 ± 0.25	1.12 ± 0.24	1.16 ± 0.24

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
