# Peer review of "Effect of Hydroxybenzoic Acids on Caffeine Detection Using Taste Sensor with Lipid/Polymer Membranes"

_sensors, 2022, doi:10.3390/s22041607_

Round 1

Reviewer 1 Report

The Authors report on a potentiometric sensor with lipid/polymer membrane sensitive to caffeine. The manuscript is written quite clearly, with many details allowing to follow the procedures. The main goal of the manuscript is to elucidate a systematic understanding of how HBAs contribute to caffeine detection. Although the previous paper, Yoshimatsu, J.; Toko, K.; Tahara, Y.; Ishida, M.; Habara, M.; Ikezaki, H.; Kojima, H.; Ikegami, S.; Yoshida, M.; Uchida, T. Development of taste sensor to detect non-charged bitter substances, Sensors (Switzerland) 2020, 20, 1–13, doi:10.3390/s20123455, gives the convincing explanation, present manuscript gives some additional valuable information and interpretation.

Detailed comments:

I have a following question to the Authors. How do you know that during soaking the lipid/polymer membrane in the modification solution containing various HBAs during quite a long time in the range from 48 to 72 hours only adsorption at the membrane surface takes place. What about ion exchange between the membrane and modification solution. Have you checked the presence of bromide ions in the modification solution after the membrane soaking? Your electrode before modification is a typical solvent/polymeric ion selective electrode with ion exchanger, which basic behavior is very well explained in a valuable book by W.E. Morf, The principles of ion-selective electrodes and of membrane transport, Akademiai Kiado, Budapest, 1981. Please, check this possibility and add the result to the manuscript.

Author Response

Response to Reviewer 1’s Comments

Dear Editor and Reviewer1,

Thank you for your comments concerning our manuscript entitled “Effect of Hydroxybenzoic Acids on Caffeine Detection Using Taste Sensor with Lipid/Polymer Membranes”. Those comments are all valuable and very helpful for revising and improving our paper, as well as the important guiding significance to our research. We have studied comments carefully and have made a correction which we hope to meet with approval. We are submitting the corrected manuscript with the suggestion incorporated in the manuscript. The main corrections in the manuscript and the responses to the reviewer’s comments are as follows:

Reviewer #1:

  1. Comment: How do you know that during soaking the lipid/polymer membrane in the modification solution containing various HBAs during quite a long time in the range from 48 to 72 hours only adsorption at the membrane surface takes place. What about ion exchange between the membrane and modification solution. Have you checked the presence of bromide ions in the modification solution after the membrane soaking? Your electrode before modification is a typical solvent/polymeric ion selective electrode with ion exchanger, which basic behavior is very well explained in a valuable book by W.E. Morf, The principles of ion-selective electrodes and of membrane transport, Akademiai Kiado, Budapest, 1981. Please, check this possibility and add the result to the manuscript.

Response- Thank you for your constructive comment. As indicated by you, the possibility of a Br- exchanger does exist in our electrode. We have mentioned in the revised manuscript (line 156-162).

To confirm this possibility, first of all, please allow me to review our previous studies on the changes to the lipid/polymer membrane caused by preconditioning, e.g., “Harada, Y.; Noda, J.; Yatabe, R.; Ikezaki, H.; Toko, K. Research on the changes to the lipid/polymer membrane used in the acidic bitterness sensor caused by preconditioning. Sensors (Basel). 2016, 16, 230, doi:10.3390/s16020230.” and “Yatabe, R.; Noda, J.; Tahara, Y.; Naito, Y.; Ikezaki, H.; Toko, K. Analysis of a lipid/polymer membrane for bitterness sensing with a preconditioning process. Sensors (Switzerland) 2015, 15, 22439–22450, doi:10.3390/s150922439”. Yatabe, R et al. have investigated the changes to lipid/polymer membrane caused by the preconditioning through three experiments (e.g., measurement of taste sensor experiment, measurement of contact angle of the membrane surface, and measurement of the amount of the bitterness substance adsorbed onto the membrane). The results for the taste sensor showed that the response to the sample and the CPA value became high and stable after 2-3 days of monosodium glutamate (MSG) solution immersion. The results of the contact angle indicated that the surface became hydrophilic and was negatively charged by the preconditioning. Harada, Y et al. have analyzed the lipid/polymer membrane for bitterness sensing with a preconditioning process by performing contact angle and surface zeta potential measurement, FTIR-RAS, measurement of taste sensor. The results for the contact angle and the surface zeta potential indicated that the surface became hydrophilic and was negatively charged by the preconditioning. The results for FTIR-RAS, XPS, and GCIB-TOF-SIMS indicated that TDAB was concentrated on the surface, and MSG was adsorbed into the membrane. Therefore, from the conclusions of these two papers, it is feasible to adsorb the modifier onto the membrane surface by immersing the membrane in the modification solution.

In the second place, please review the CPA results on the manuscript again (e.g., Figure 7(c), 8(c), and 9(c) in the manuscript). These CPA results confirmed that there was caffeine adsorbed on the membrane surface during the caffeine detection. Since caffeine is a non-charged substance and the log D of caffeine is -0.44, the adsorption of caffeine does not lie on the electrostatic or hydrophobic interaction, but the intermolecular hydrogen bonds between caffeine and HBAs adsorbed on the membrane. In other words, the CPA values demonstrated that there is the adsorption of HBAs on the membrane surface. Furthermore, the presence of CPA values demonstrates the feasibility of surface modification by immersing the membranes in HBAs solution.

Thirdly, because Br- and HBAs have the same negative charge, an ion exchanger can take place on the membrane surface, as the reviewer pointed out. However, the difference between Br- and HBAs is their hydrophobicity. Because of the overwhelming hydrophobicity of HBA, HBAs can be adsorbed onto the membrane surface by hydrophobic interaction. Therefore, Vr is shifted to the negative side compared to Vr obtained without surface modification owing to the adsorption of HBAs.

Moreover, on page 11 of the reference book that you provided, there is a description of the ion exchanger. It is stated in the text that liquid-membrane electrodes with electrically charged ion-exchange sites generally show permselectivity for oppositely charged counterions, which means that liquid-membrane electrodes with TDAB membranes have a strong electrostatic interaction on negatively charged ions. In the case of nearly complete dissociation between sites and counterions, the selectivity between different counterions of the same negative charge is dictated mainly by the extraction behavior of the solvating membrane medium. The following monotonic selectivity sequence is obtained for membrane electrodes based on dissociated antlion-exchangers.

R- > ClO4- > I- > NO3- ~ Br- > Cl- > F-

Here, ionized HBA is R-. From the monotonic selectivity sequence, a Br- exchanger does exist, but it is not the main factor that makes the decrease in Vr. Therefore, ionized HBAs can be adsorbed on the membrane surface owing to the hydrophobic interaction, causing the negative charge of ionized HBAs to neutralize the positive charge of TDAB, resulting in a negative shift of Vr.

Finally, we have tried to mix HBAs directly into the membrane. Taking 2,6-DHBA as an example, please take a look at Figure R-1 in the uploaded response letter. As shown in Figure R-1, the responses to caffeine are not much different from those obtained with the surface modification (Figure 7(b) in the manuscript). Therefore, we finally chose the method of immersing the lipid membrane in HBAs solution to complete the surface modification of the membrane.

We tried our best to improve the manuscript and made some changes in the manuscript. We appreciate for Editors/Reviewers’ warm work earnestly, and hope that the correction will meet with approval.

Once again, thank you very much for your comments and suggestions.

Reviewer 2 Report

General Comments: This paper explores the modification of lipid membranes for detecting caffeine. I find it interesting to read and for future applications also. Data supports the hypothesis well. This paper may be accepted with minor corrections as per the following specific comments:

Specific Comments

  1. Line 144: What does "CPA" represent in the text? Expand it once.
  2. Line 186-200: Please mention the concentration of modification solutions. Same concentration or different concentration for all HBAs.
  3. Line 205-214: Please mention the concentration of modification solutions.
  4. Line 214: Figure 6 caption: Please indicate n=? for error bars.
  5. Line 183 and Table 2 states that 6 different HBAs were studied for modification of lipid-polymer membrane, however the data in Section 3.2 gives data for only three selected HBA (2,6 DHBA, 2-HBA, 3,5 DHBA) for reasons not mentioned in the article. Further, if we look at modified sensor for 3,5 DHBA in Figure 5 and 6, it is the least sensitive sensor. I failed to find a suitable reason for this selection.
  6. Line 241: What is "E"?
  7. ‌Line 249-250: In section 3.2.2, it was stated that CPA value increased with increasing concentration, but Figure 8(c) is indicating that 0.3 mM TDAB is decreasing in between 0.01 to 1 wt% concentration. Justify this phenomenon.
  8. ‌Line 263: It states that Vr increased with increasing concentration, but Figure 9(a) is indicating that 0.3 mM TDAB is not significantly increasing with increasing concentration and even decreases or have negligible effect at one point. Justify this happening.
  9. Line 274-277: The comparisons of Vr may be performed with adequate statistical tests to find significant outcomes and give strong evidence for obtained results.

Author Response

Response to Reviewer 2’s Comments

Dear Editor and Reviewer 2,

Thank you for the comments concerning our manuscript entitled “Effect of Hydroxybenzoic Acids on Caffeine Detection Using Taste Sensor with Lipid/Polymer Membranes”. Those comments are all valuable and very helpful for revising and improving our paper, as well as the important guiding significance to our research. We have studied comments carefully and have made corrections which we hope to meet with approval. We are submitting the corrected manuscript with the suggestion incorporated in the manuscript.  The main corrections in the manuscript and the responses to the reviewer’s comments are as follows:

Reviewer #2:

  1. Line 144: What does "CPA" represent in the text? Expand it once.

Response- Thank you so much for your minute observation and valuable comments. We have expanded it in the revised manuscript (line 144).

  1. Line 186-200: Please mention the concentration of modification solutions. Same concentration or different concentration for all HBAs.

Response- Thank you so much for your valuable suggestion. We have mentioned it in the revised manuscript (line 195-196).

  1. Line 205-214: Please mention the concentration of modification solutions.

Response- Thank you very much for your valuable suggestion. We have mentioned it in the revised manuscript (line 215-216). Hopefully, you will find it justified.

  1. Line 214: Figure 6 caption: Please indicate n=? for error bars.

Response- Thank you so much for your comment. We have added it during the revision of the manuscript (line 223-224).

  1. Line 183 and Table 2 states that 6 different HBAs were studied for modification of lipid-polymer membrane, however the data in Section 3.2 gives data for only three selected HBA (2,6 DHBA, 2-HBA, 3,5 DHBA) for reasons not mentioned in the article. Further, if we look at modified sensor for 3,5 DHBA in Figure 5 and 6, it is the least sensitive sensor. I failed to find a suitable reason for this selection.

Response- Thank you so much for your comments. We have added the reason it in the revised manuscript (line 227-232, line 258-260, and line 276-277). We have chosen these three types of HBAs for three reasons. The first reason is that in our previous study (Yoshimatsu, J.; Toko, K.; Tahara, Y.; Ishida, M.; Habara, M.; Ikezaki, H.; Kojima, H.; Ikegami, S.; Yoshida, M.; Uchida, T. Development of taste sensor to detect non-charged bitter substances, Sensors (Switzerland) 2020, 20, 1–13, doi:10.3390/s20123455), the sensitivity of taste sensors to caffeine detection had been related to the number of intramolecular hydrogen bonds of HBAs. Therefore, we chose 2,6-DHBA with two intramolecular hydrogen bonds, 2-HBA with one intramolecular hydrogen bond, and 3,5-DHBA without intramolecular hydrogen bonds. The second reason is that the membrane modified with 2,6-DHBA responds best to caffeine. Comparing 2,6-DHBA with other HBAs, the best response conditions for caffeine detection could be derived. The third reason is that in Yoshimatsu, J's paper, 2-HBA and 3,5-DHBA have not been studied yet. Therefore, in the manuscript, we chose 2,6-DHBA, 2-HBA, and 3,5-DHBA to investigate the effects of HBAs on caffeine detection.

  1. Line 241: What is "E"?

Response- Thank you so much for your comments. Here “E” refers to an abbreviation for “Effect”. We have mentioned it in the revised manuscript (line 235-236). This abbreviation is used to describe the concentration-effect curves (e.g., E/[HBAs] curves, [HBAs] is the concentration of HBAs). “E/[HBAs] curves” is referenced from the paper, Giraldo, J.; Vivas, N.M.; Vila, E.; Badia, A. Assessing the (a)symmetry of concentration-effect curves: Empirical versus mechanistic models. Pharmacol. Ther. 2002, 95, 21–45, doi:10.1016/S0163-7258(02)00223-1.

  1. Line 249-250: In section 3.2.2, it was stated that CPA value increased with increasing concentration, but Figure 8(c) is indicating that 0.3 mM TDAB is decreasing in between 0.01 to 1 wt% concentration. Justify this phenomenon.

Response- Thank you so much for your suggestion. Yes, you are right. We have added this justified phenomenon in the revised version of the manuscript (line 269-272).

  1. Line 263: It states that Vr increased with increasing concentration, but Figure 9(a) is indicating that 0.3 mM TDAB is not significantly increasing with increasing concentration and even decreases or have negligible effect at one point. Justify this happening.

Response- Thank you so much for your suggestion. As you mentioned, there is a decrease in Vr for 0.3 mM TDAB membrane. We have justified this happening in the revised version of the manuscript (line 279-282).

  1. Line 274-277: The comparisons of Vr may be performed with adequate statistical tests to find significant outcomes and give strong evidence for obtained results.

Response- Thank you so much for your comments. The sentence about the comparisons of Vr is restructured to make it more clear for the reader. We have added the restructured sentence in the revised manuscript (line 302-307).

We tried our best to improve the manuscript and made some changes in the manuscript. We appreciate for Editors/Reviewers’ warm work earnestly and hope that the correction will meet with approval.

Once again, thank you very much for your comments and suggestions.

Round 2

Reviewer 1 Report

All my comments and questions are carrefully taken into account and carrefully explained and answered.